

# Association between passing return-to-sport testing and re-injury risk in patients after anterior cruciate ligament reconstruction surgery: a systematic review and meta-analysis

Wenqi Zhou, Xihui Liu, Qiaomei Hong, Jingping Wang and Xiaobing Luo

Department of Sport Medicine, Sichuan Orthopedic Hospital, Chengdu, Sichuan, China

## ABSTRACT

**Background**. Inconsistent results have been obtained regarding the association between return-to-sport (RTS) testing and the risk of subsequent re-injury following anterior cruciate ligament reconstruction (ACLR). We therefore conducted a systematic review and meta-analysis to assess the potential association between passing of RTS and the risk of re-injury for patients after ACLR.

**Methods**. This meta-analysis was registered in INPLASY with the registration number INPLASY202360027. The electronic databases MedLine, EmBase, and the Cochrane library were systematically searched to identify eligible studies from their inception up to September 2023. The investigated outcomes included knee injury, secondary ACL, contralateral ACL injury, and graft rupture. The pooled odds ratios (ORs) and 95% confidence intervals (CIs) were calculated using the random-effects model.

**Results**. A total number of nine studies involving 1410 individuals were selected for the final quantitative analysis. We noted that passing RTS test was not associated with the risk of subsequent knee injury (OR: 0.95; 95% CI: 0.28–3.21; $P = 0.929$), secondary ACL injury (OR: 0.98; 95% CI: 0.55–1.75; $P = 0.945$), and contralateral ACL injury (OR: 1.53; 95% CI: 0.63–3.71; $P = 0.347$). However, the risk of graft rupture was significantly reduced (OR: 0.49; 95% CI: 0.33–0.75; $P = 0.001$).

**Conclusions**. This study found that passing RTS test was not associated with the risk of subsequent knee injury, secondary ACL injury, and contralateral ACL injury, while it was associated with a lower risk of graft rupture. Thus, it is recommended that patients after ACLR pass an RTS test in clinical settings.

Corresponding author
Xiaobing Luo, luoxiaobing1969@126.com

## INTRODUCTION

Pivoting, cutting, or jumping sports (*e.g.*, basketball, soccer, or team handball) pose an increased risk of anterior cruciate ligament (ACL) injury for athletes (*Chen et al., 2022*; *Reist et al., 2023*). An earlier study already revealed that female athletes were more susceptible to ACL injuries; the incidence rate in female soccer and basketball players (approximately

5%) was three times higher than that in males (*Prodromos et al., 2007*). Athletes usually undergo surgical reconstruction to restore knee stability and function, enabling them to achieve the goal of return to sports after ACL injury after rehabilitation (*Keays et al., 2022*; *Swirtun, Eriksson & Renström, 2006*; *Tashman, Kopf & Fu, 2008*). However, an ACL rupture can have a significant impact on future sports participation and may even mark the end of a promising career (*Tashman, Kopf & Fu, 2008*) with an high risk of re-injury (*Rodriguez-Merchan & Valentino, 2022*). Therefore, it is necessary to assess the subsequent re-injury risk of athletes after an ACL injury to improve their career prospects (*Ardern et al., 2016*).

ACL reconstruction (ACLR) followed by rehabilitation is considered the gold standard treatment strategy for athletes after an ACL injury, with the ultimate goal of returning to sport (RTS) (*Paschos & Howell, 2016*). To ensure a safe and successful return to sport activities, specific criteria have been developed to assess athletes' readiness to return to their pre-injury performance level and minimize the risk of re-injury. This decision-making process incorporates various multidimensional aspects (*Bien & Dubuque, 2015*; *Buckthorpe, 2021*; *Creighton et al., 2010*; *Grindem et al., 2016*). Currently, RTS testing primarily focuses on evaluating the restoration of functional and neuromuscular levels through assessments, such as quadriceps strength tests, single-legged hop tests, and self-report questionnaires (*Ardern et al., 2016*; *Buckthorpe, 2021*; *Di Stasi et al., 2013*). However, studies examining the role of RTS testing in predicting subsequent re-injury after ACL reconstruction have produced inconsistent results (*Graziano et al., 2017*; *Grindem et al., 2016*; *Kyritsis et al., 2016*; *Nawasreh et al., 2017*; *Sousa et al., 2017*; *Wellsandt, Failla & Snyder-Mackler, 2017*). Therefore, it is crucial to clarify the significance of RTS testing in re-injury for athletes after ACLR, as its impact remains inconclusive. In this study, we conducted a comprehensive analysis of published studies to determine the association between RTS testing and knee injury, secondary ACL injury, contralateral ACL injury, or graft rupture. Our review and analysis of the available evidence provides a clearer understanding of the role of RTS testing in the prognosis of athletes after ACL injury.

## METHODS

### Data sources, search strategy, and selection criteria

This review was conducted and reported according to the Preferred Reporting Items for Systematic Reviews and Meta-Analysis Statement (*Page et al., 2021*). Studies that examined the impact of the RTS test on the risk of subsequent re-injury for athletes after ACLR were eligible for inclusion in our analysis, with no restrictions on the publication language. We searched electronic databases (MedLine, EmBase, and Cochrane Library) for articles published from the inception of the databases to September 2023. The following core search terms were used: ''anterior cruciate ligament reconstruction'' OR ''ACL reconstruction'' AND ''return to sport'' OR ''return to sport criteria'' OR ''return to play'' OR ''return to play criteria'' OR ''functional testing'' OR ''return to athletic*''. The details of the search strategy are presented in Supplemental Material 1. Additionally, we manually searched the reference lists of relevant articles to identify additional eligible studies. The study selection

was conducted based on the medical subject heading, methods, patient population, design, exposure, and outcome variables.

The literature search and study selection process were independently conducted by two authors (Wenqi Zhou and Qiaomei Hong) using a standardized approach. Disagreements were resolved by the first author until a consensus was reached. The inclusion criteria were based on the following predefined eligibility criteria: (1) Participants: post-ACLR athletes; (2) Exposure: passing RTS testing; (3) Control: failure in passing RTS testing; (4) Outcomes: knee injury (all knee injuries and ACL injury), secondary ACL injury (defined as contralateral ACL injury and graft rupture), contralateral ACL injury, or graft rupture; and (5) Study design: no restrictions were placed on study design, and prospective and retrospective design were eligible. Detailed methodology can be referred to previous words (*Chen et al., 2023*; *Li et al., 2021*; *Zhang et al., 2021a*; *Zhang et al., 2021b*). Animal experiments, reviews, and case reports were excluded as they could not provide sufficient data.

### Data collection and quality assessment

The data collected included the first author's name, publication year, study design, country, sample size, male percentage (%), mean age (years), physical condition of the participants, RTS test battery, RTS test time and pass rate, and reported outcomes. The methodological quality of each individual study in the meta-analysis was assessed using the Newcastle-Ottawa Scale (NOS), which consists of selection (four items), comparability (one item), and outcome (three items) criteria (*Wells et al., 2009*). A "star system" ranging from 0 to 9 was used to evaluate each study. These assessments were independently performed by two authors (Wenqi Zhou and Qiaomei Hong), and any disagreements were resolved through a review of the full-text of the original articles by an additional author (Xiaobing Luo), followed by a discussion (*Dutaillis et al., 2003*; *Li et al., 2022*; *Zhang et al., 2021a*; *Zhang et al., 2021b*).

### Statistical analysis

We examined the association of RTS testing with the risk of knee injury, secondary ACL injury, contralateral ACL injury, and graft rupture considering the incidence rate and the sample size in each individual study. Then, the pooled odds ratios (ORs) and 95% confidence intervals (CIs) were calculated using the random-effects model (*Ades, Lu & Higgins, 2005*; *DerSimonian & Laird, 1986*). The heterogeneity across the studies was analyzed using the $I^2$ and Q statistic; the significance of heterogeneity was determined using $I^2 > 50\%$ or $P < 0.10$ (*Higgins et al., 2003*; *Deeks, Higgins & Altman, 2008*). The robustness of the pooled conclusions was assessed through a sensitivity analysis (*Tobias, 1999*). Subgroup analysis was also conducted based on study design, country, percentage male, mean age, and study quality. The differences between subgroups were compared using the interaction $P$ test. The publication bias for each investigated outcome was assessed through visual inspections of funnel plots, and quantitatively assessed using the Egger's and Begg's tests (*Begg & Mazumdar, 1994*; *Egger et al., 1997*). All reported $P$- values are two-sided; $P < 0.05$ was considered to indicate statistical significance. Statistical analyses were performed using STATA software (version 12.0; Stata Corporation, College Station, TX, USA).

## RESULTS

### Literature search

A total number of 1,591 articles were identified in our initial electronic search, 1,527 of which were excluded due to duplication and not fitting the inclusion criteria. The remaining 64 studies were retrieved for further full-text evaluations, of which 55 were excluded for the following reasons: inappropriate control ($n = 26$), insufficient data ($n = 21$), and reviews ($n = 8$). Further, nine studies were selected for the final meta-analysis (*Graziano et al., 2017*; *Grindem et al., 2016*; *Kyritsis et al., 2016*; *Nawasreh et al., 2017*; *Paterno et al., 2022*; *Raoul et al., 2019*; *Sousa et al., 2017*; *Webster & Feller, 2019*; *Wellsandt, Failla & Snyder-Mackler, 2017*). The manual search for reference lists yielded eight potentially relevant articles, but no eligible study was detected after detailed evaluations (Fig. 1). The baseline characteristics of the included studies and individuals are presented in Table 1.

### Study characteristics

Of the nine included studies (a total number of 1,410 individuals), seven were prospective cohort studies, and the remaining two were retrospective studies. The mean age range of the athletes was 12.0–28.4 years, and 42–329 athletes were included in each individual study. One study included only male athletes, whereas the remaining eight studies included both male and female athletes. Five studies were conducted in USA, whereas the remaining four studies were performed in Europe or Australia. Study quality was assessed by using the NOS scale. Three of the studies obtained eight stars, three studies seven stars, and the remaining three studies received six stars.

### Knee injury

A total number of six studies reported the association between passing RTS test and the risk of knee injury. RTS test passing by athletes was not associated with the risk of knee injury as compared with individuals failing the RTS test (OR: 0.95; 95% CI [0.28–3.21]; $P = 0.929$; Fig. 2). Significant heterogeneity was detected ($I^2 = 82.6\%$; $P < 0.001$). Our sensitivity analysis found the pooled conclusion was robustness and not altered by sequential excluding individual study (Supplemental Material 2). For pooled studies conducted in Europe, the subgroup analysis found that passing the RTS test was associated with a lower risk of knee injury. However, it was associated with an increased risk of knee injury in pooled studies performed in the USA or in studies with a lower quality (Table 2). The interaction test results showed that the passing of the RTS test with the risk of knee injury could affect by study design, country, percentage male, and study quality. There was no significant publication bias for knee injury (*P*- value for the Egger's test: 0.558; *P*- value for the Begg's test: 0.452; Supplemental Material 3).

### Secondary ACL injury

After pooling all included studies, we found that passing a RTS test by athletes did not affect their risk of secondary ACL injury (OR: 0.98; 95% CI [0.55–1.75]; $P = 0.945$; Fig. 3). A significant heterogeneity among the included studies was detected ($I^2 = 53.1\%$; $P = 0.030$). Thus, sensitivity analysis was conducted for secondary ACL injury. Our findings were not

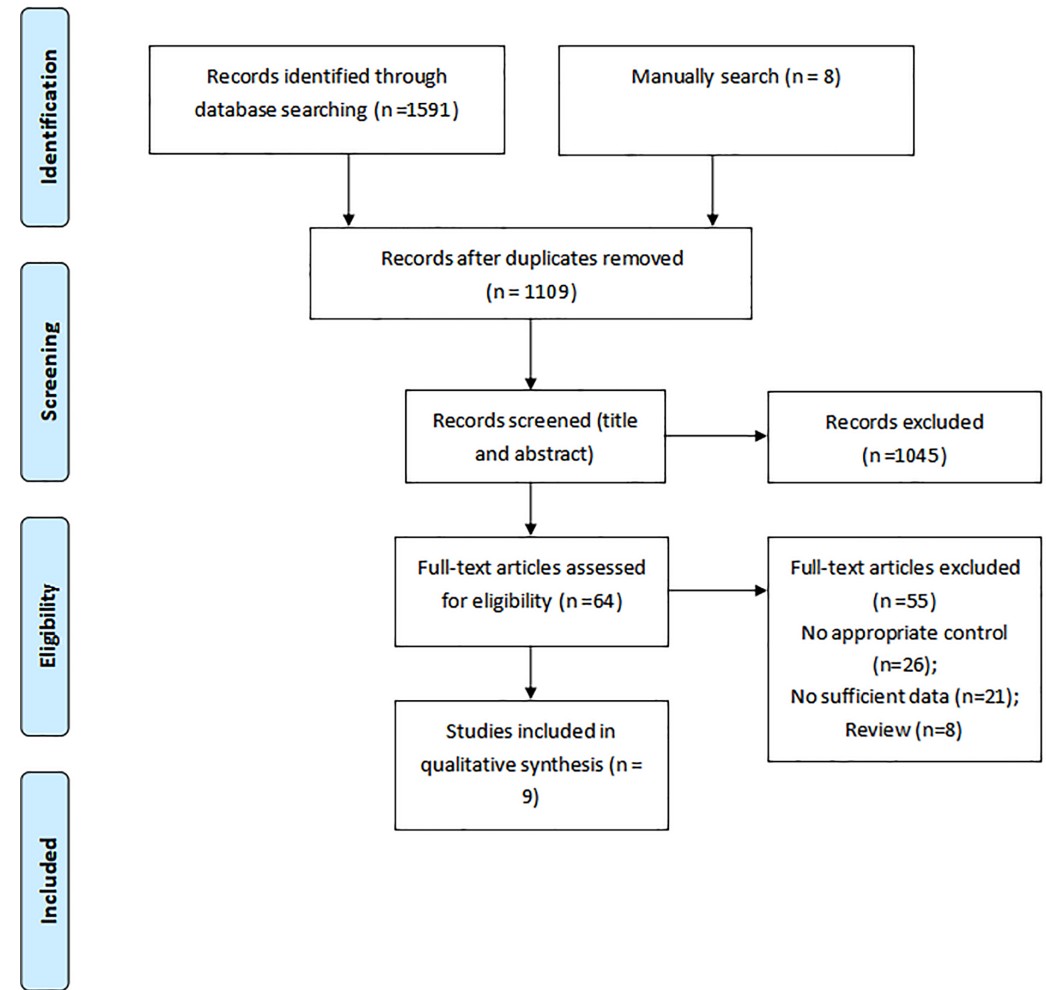

**Figure 1    Flow diagram of the literature search and studies selection process.**

affected by the exclusion of any specific study from the pooled analysis (Supplemental Material 2). The results of subgroup analyses of pooled studies conducted in Europe or Australia showed that passing an RTS test was associated with a reduced risk of secondary ACL injury, whereas an increased risk was detected in the pooled studies performed in the USA. Additionally, study design, country, percentage male, and study quality could affect the role of passing RTS with the risk of secondary ACL injury (Table 2). No significant publication bias for secondary ACL injury was detected (*P*- value for Egger's test: 0.772; *P*-value for Begg's test: 0.917; Supplemental Material 3).

## Contralateral ACL injury

A total number of eight studies reported the association between passing RTS test with the risk of contralateral ACL injury. Athletes passing RTS test was not associated with the risk of contralateral ACL injury (OR: 1.53; 95% CI [0.63–3.71]; $P = 0.347$; Fig. 4). Significant

**Table 1  The baseline characteristics of identified studies and patients.**

| Study | Study design | Country | Sample size | Percent of male (%) | Mean age (years) | Participants | RTS test battery | RTS test time and pass rate | Study quality |
|---|---|---|---|---|---|---|---|---|---|
| *Grindem et al. (2016)* | Prospective | Norway | 100 | 46.0 | 24.3 | Level I or II sports 67% HS; 33% PT | Two self-report (KOS-ADLS; global rating scale), quadriceps strength, 4 hop tests (distance, timed, triple hop and triple crossover); LSI ≥ 90 to pass | Between 3 and 23 months; 24% (18/74) pass rate for those who RTS | 7 |
| *Kyritsis et al. (2016)* | Prospective | Belgium | 158 | 100.0 | 21.5 | HS (68%) PT (32%) | Quadriceps strength, 3 hop tests (single, triple, triple crossover), on field rehabilitation, running *t*-test; Quadriceps deficit <10%, LSI >90 (single, triple, crossover hop, running t test <11 s; completed on field sport specific rehabilitation | Unclear if all measured before RTS; pass rate of 73% (116/158) | 8 |
| *Nawasreh et al. (2017)* | Prospective | USA | 95 | 66.3 | 27.2 | Level I/II sports, 38% autograft, 62% allograft | Two self-report (KOS-ADLS; global rating scale), quadriceps strength, 4 hop tests (distance, timed, triple hop and triple crossover); LSI ≥ 90 to pass | 6 months; (48/95) pass rate 81% (30/37) who passed RTS tests at 6 months returned at 12 months; 44% (19/43) who failed RTS tests at 6 months returned at 12 months; 84% (27/32) who passed RTS tests at 6 months returned at 24 months; 46% (13/28) who failed RTS tests at 6 months returned at 24 months | 7 |

Zhou et al. (2024), *PeerJ*, DOI 10.7717/peerj.17279

**Table 1** (*continued*)

| Study | Study design | Country | Sample size | Percent of male (%) | Mean age (years) | Participants | RTS test battery | RTS test time and pass rate | Study quality |
|---|---|---|---|---|---|---|---|---|---|
| *Sousa et al. (2017)* | Retrospective | USA | 223 | 41.3 | 22.0 | Median Tegner = 6 (2-10); PT autograft (59%) PT allograft (28%) HS (13%) | Strength and three functional tests (vertical jump, single hop, triple jump); LSI ≥ 85 for strength and ≥ 90 function. Overall pass if pass 6 of 7 tests | 6 months; 23% (52/223) pass rate | 6 |
| *Wellsandt, Failla & Snyder-Mackler (2017)* | Prospective | USA | 70 | 67.1 | 26.6 | Cutting and pivot sports; HS (40%) soft tissue allograft (60%) | Quadriceps strength, 4 hop tests (distance, timed, triple hop and triple crossover); LSI ≥ 90 to pass; EPIC ≥ 90 to pass (using pre-surgery uninvolved limb as comparator) | 6 months; 57% (40/70) LSI; 29% (20/70) EPIC | 6 |
| *Graziano et al. (2017)* | Retrospective | USA | 42 | 71.4 | 12.0 | Various sports | Stability, strength, hop for distance; Pass cut-off not reported | From 5 months, unclear if all measured before RTS; pass rate of 90% (37/41); One patient had playground accident at 3 months and reinjured knee | 6 |
| *Raoul et al. (2019)* | Prospective | France | 234 | 73.9 | 28.4 | HS (82.9%), fascia lata (10.7%) and PT (6.4%) | Functional performance of the knee by isokinetic tests performed on a dynamometer to measure quadriceps and hamstring strength, and neuromuscular assessment based on single-leg hop tests. | 6.5 months; 18.8% pass rate | 7 |

Zhou et al. (2024), *PeerJ*, DOI 10.7717/peerj.17279

**Table 1** (*continued*)

| Study | Study design | Country | Sample size | Percent of male (%) | Mean age (years) | Participants | RTS test battery | RTS test time and pass rate | Study quality |
|---|---|---|---|---|---|---|---|---|---|
| *Webster & Feller (2019)* | Prospective | Australia | 329 | 60.8 | 17.2 | Various sports | Range of knee motion (passive flexion and extension deficits), instrumented anterior knee laxity, and single- and triple-crossover hop for distance | 12 months; 28.4% pass rate | 8 |
| *Paterno et al. (2022)* | Prospective | USA | 159 | 29.6 | 17.2 | HS (54.1%), PT (37.1%), and allograft (8.8%) | All 6 RTS tests at a criterion level of 90% (or 90 of 100) limb symmetry | 7 months; 26% pass rate | 8 |

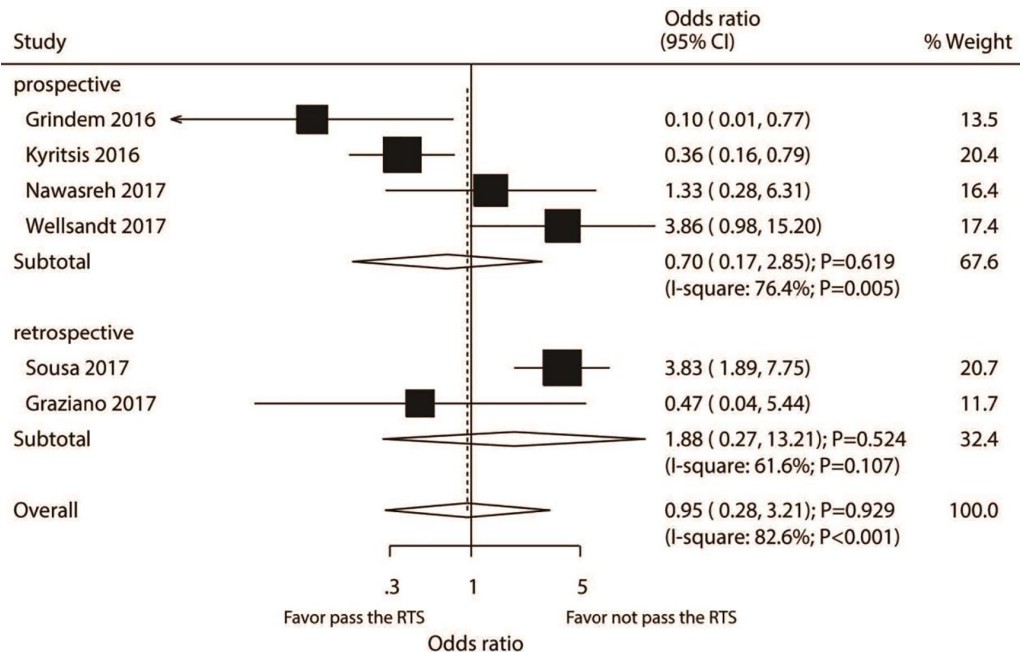

**Figure 2** Passing RTS test and the risk of knee injury. (*Grindem et al., 2016*; *Kyritsis et al., 2016*; *Nawasreh et al., 2017*; *Wellsandt, Failla & Snyder-Mackler, 2017*; *Sousa et al., 2017*; *Graziano et al., 2017*).

heterogeneity was observed across the included studies ($I^2 = 46.0\%$; $P = 0.073$). Sensitivity analysis found that passing an RTS test might be associated with an elevated risk of contralateral ACL injury (Supplemental Material 2). Moreover, subgroup analysis results suggested that passing an RTS test was associated with an increased risk of contralateral ACL injury c (Table 2). There was no significant publication bias for contralateral ACL injury (*P*- value for Egger's test: 0.550; *P*- value for Begg's test: 0.711; Supplemental Material 3).

## Graft rupture

A total number of eight studies established the existence of an association between passing an RTS test and an increase in the risk of graft rupture. Passing an RTS test was associated with a reduced risk of graft rupture in athletes (OR: 0.49; 95% CI [0.33–0.75]; $P = 0.001$; Fig. 5). We found no evidence of heterogeneity across the included studies ($I^2 = 0.0\%$; $P = 0.430$). The pooled conclusion was validated by the sequential exclusion of individual studies (Supplemental Material 2). The subgroup analysis results of pooled studies with a prospective design, studies conducted in Europe or Australia, those with a male proportion ≥ 60%, with a mean age <25.0 years, and pooled studies with high quality revealed that passing an RTS test was associated with a reduced risk of graft rupture in athletes (Table 2). No significant publication bias for graft rupture was detected (*P*- value for Egger's test: 0.944; *P*- value for Begg's test: 0.711; Supplemental Material 3).

**Table 2  Subgroup analyses for investigated outcomes.**

| Outcomes | Factors | Subgroup | OR and 95%CI | P value | I² (%)/P value | P value between subgroups |
|---|---|---|---|---|---|---|
| Knee injury | Study design | Prospective | 0.70 (0.17–2.85) | 0.619 | 76.4/0.005 | <0.001 |
| | | Retrospective | 1.88 (0.27–13.21) | 0.524 | 61.6/0.107 | |
| | Country | Europe or Australia | 0.26 (0.08–0.83) | 0.023 | 31.8/0.226 | <0.001 |
| | | USA | 2.68 (1.30–5.53) | 0.007 | 21.8/0.280 | |
| | Percentage male (%) | ≥ 60.0 | 0.97 (0.27–3.42) | 0.957 | 68.9/0.022 | 0.011 |
| | | <60.0 | 0.68 (0.01–34.11) | 0.846 | 92.1/<0.001 | |
| | Mean age (years) | ≥ 25.0 | 2.42 (0.86–6.82) | 0.094 | 1.3/0.314 | 0.183 |
| | | <25.0 | 0.57 (0.10–3.28) | 0.532 | 88.5/<0.001 | |
| | Study quality | High | 0.40 (0.12–1.31) | 0.131 | 53.1/0.118 | <0.001 |
| | | Low | 3.09 (1.36–7.04) | 0.007 | 24.5/0.266 | |
| Secondary ACL injury | Study design | Prospective | 0.85 (0.47–1.51) | 0.571 | 42.4/0.108 | 0.034 |
| | | Retrospective | 1.40 (0.23–8.38) | 0.714 | 53.5/0.142 | |
| | Country | Europe or Australia | 0.52 (0.32–0.86) | 0.011 | 0.0/0.524 | <0.001 |
| | | USA | 1.85 (1.12–3.04) | 0.016 | 0.0/0.627 | |
| | Percentage male (%) | ≥ 60.0 | 0.70 (0.40–1.24) | 0.220 | 22.1/0.268 | 0.008 |
| | | <60.0 | 1.61 (0.71–3.69) | 0.257 | 43.9/0.168 | |
| | Mean age (years) | ≥ 25.0 | 1.52 (0.57–4.09) | 0.403 | 0.0/0.623 | 0.354 |
| | | <25.0 | 0.86 (0.42–1.79) | 0.693 | 67.2/0.009 | |
| | Study quality | High | 0.75 (0.41–1.35) | 0.330 | 39.7/0.141 | 0.010 |
| | | Low | 2.10 (0.99–4.46) | 0.053 | 7.2/0.340 | |
| Contralateral ACL injury | Study design | Prospective | 0.89 (0.43–1.82) | 0.740 | 6.5/0.375 | 0.012 |
| | | Retrospective | 3.07 (0.75–12.61) | 0.120 | 19.9/0.264 | |
| | Country | Europe or Australia | 0.71 (0.36–1.40) | 0.326 | 0.0/0.392 | 0.006 |
| | | USA | 3.37 (1.41–8.05) | 0.006 | 0.0/0.514 | |
| | Percentage male (%) | ≥ 60.0 | 0.75 (0.39–1.42) | 0.374 | 0.0/0.489 | 0.004 |
| | | <60.0 | 4.12 (1.59–10.68) | 0.003 | 0.0/0.847 | |
| | Mean age (years) | ≥ 25.0 | 1.82 (0.34–9.92) | 0.487 | 0.0/0.502 | 0.859 |
| | | <25.0 | 1.48 (0.47–4.66) | 0.505 | 65.4/0.021 | |
| | Study quality | High | 0.72 (0.38–1.40) | 0.338 | 0.0/0.551 | 0.004 |
| | | Low | 3.85 (1.54–9.61) | 0.004 | 0.0/0.480 | |
| Graft rupture | Study design | Prospective | 0.49 (0.28–0.84) | 0.010 | 16.6/0.307 | 1.000 |
| | | Retrospective | 0.54 (0.14–2.12) | 0.381 | 0.0/0.319 | |
| | Country | Europe or Australia | 0.40 (0.23–0.71) | 0.002 | 13.7/0.324 | 0.191 |
| | | USA | 0.85 (0.34–2.08) | 0.714 | 0.0/0.615 | |
| | Percentage male (%) | ≥ 60.0 | 0.49 (0.28–0.84) | 0.009 | 16.4/0.308 | 1.000 |
| | | <60.0 | 0.54 (0.12–2.35) | 0.410 | 6.8/0.300 | |
| | Mean age (years) | ≥ 25.0 | 1.04 (0.34–3.15) | 0.945 | 0.0/0.800 | 0.158 |
| | | <25.0 | 0.42 (0.25–0.71) | 0.001 | 12.1/0.337 | |
| | Study quality | High | 0.44 (0.24–0.82) | 0.010 | 22.3/0.272 | 0.467 |
| | | Low | 0.71 (0.25–1.98) | 0.507 | 0.0/0.520 | |

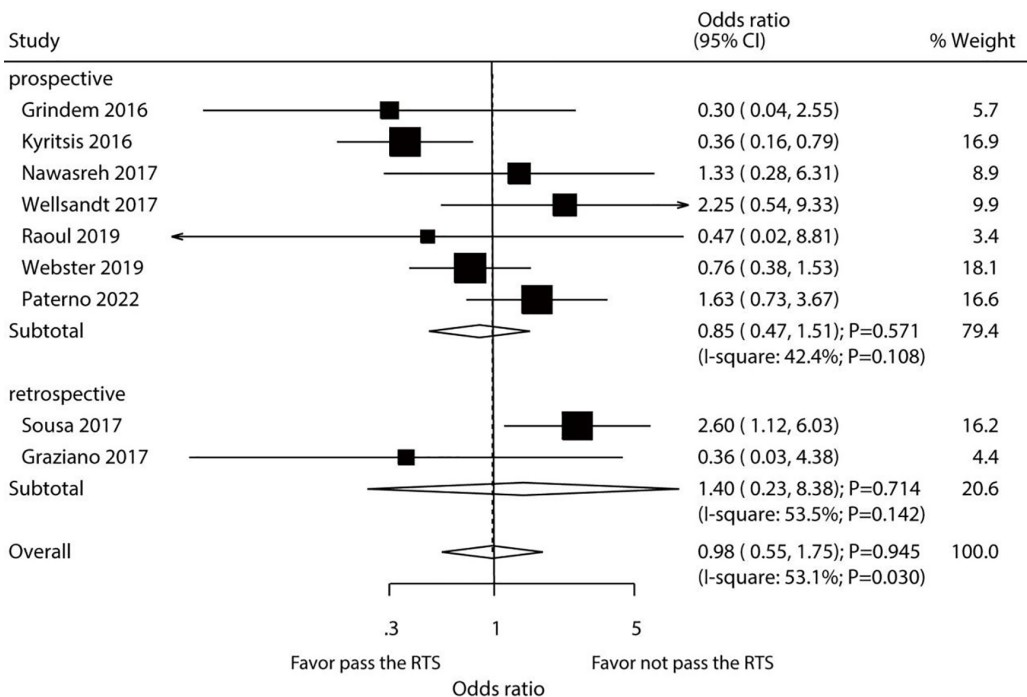

**Figure 3** **Passing RTS test and the risk of secondary ACL injury.** (*Grindem et al., 2016*; *Kyritsis et al., 2016*; *Nawasreh et al., 2017*; *Wellsandt, Failla & Snyder-Mackler, 2017*; *Raoul et al., 2019*; *Webster & Feller, 2019*; *Paterno et al., 2022*; *Sousa et al., 2017*; *Graziano et al., 2017*).

## DISCUSSION

The present study analyzed the evidence reported in previously published studies and explored the correlations between passing the RTS test and the likelihood of sustaining an knee injury, secondary ACL injury, contralateral ACL injury, or graft rupture. This comprehensive quantitative analysis assessed the data of 1,410 individuals across seven prospective cohort studies and two retrospective studies, encompassing a broad range of population characteristics. The findings from our meta-analysis suggest that passing an RTS test is not correlated with the incidence of knee injury, secondary ACL injury, and contralateral ACL injury. Our findings are consistent with those of a study conducted by *Welling et al. (2020)*, which reported that passing RTS tests after ACL reconstruction was associated with a greater likelihood for return to sport but failed to identify secondary injury risk. Moreover, passing an RTS test by athletes is associated with a reduced risk of graft rupture. In addition, the role of passing an RTS test may be influenced by the specific study design, country, percentage of males included, and the study quality.

Several systematic reviews and meta-analyses have already highlighted the potential role of the RTS test in assessing the prognosis of ACL injury. An analysis of four studies, conducted by *Losciale et al. (2019)* found that passing an RTS test was not statistically significantly associated with a risk of a secondary ACL injury. Another systematic review and meta-analysis performed by *Webster & Feller (2019)* included 17 studies. The authors

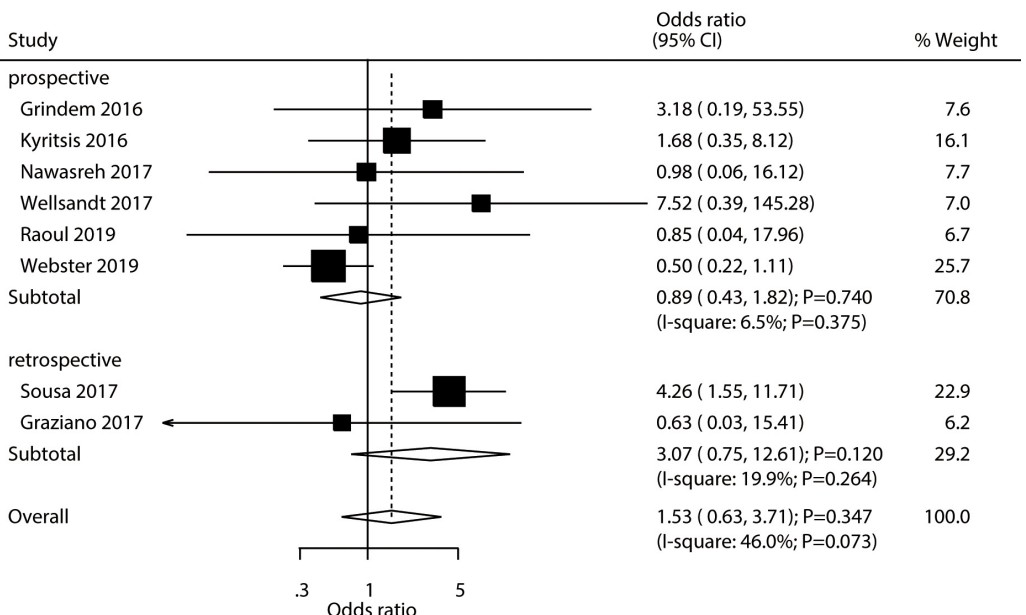

**Figure 4** **Passing RTS test and the risk of contralateral ACL injury.** (*Grindem et al., 2016*; *Kyritsis et al., 2016*; *Nawasreh et al., 2017*; *Wellsandt, Failla & Snyder-Mackler, 2017*; *Raoul et al., 2019*; *Webster & Feller, 2019*; *Sousa et al., 2017*; *Graziano et al., 2017*).

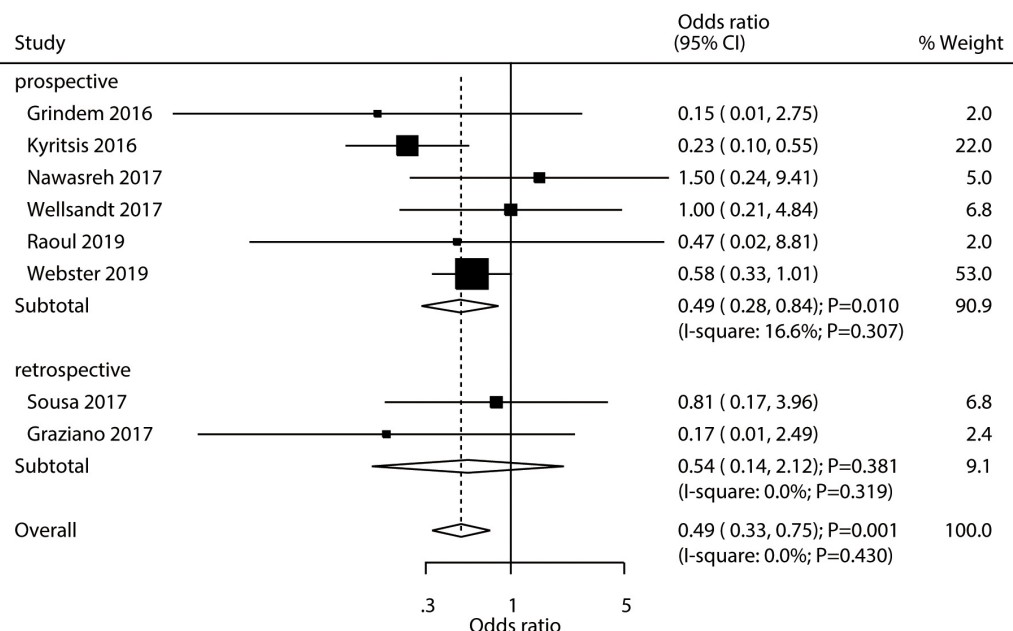

**Figure 5** **Passing RTS test and the risk of graft rupture.** (*Grindem et al., 2016*; *Kyritsis et al., 2016*; *Nawasreh et al., 2017*; *Wellsandt, Failla & Snyder-Mackler, 2017*; *Raoul et al., 2019*; *Webster & Feller, 2019*; *Sousa et al., 2017*; *Graziano et al., 2017*).

reported equivocal findings regarding the validity of current RTS test batteries in reducing the risk of graft rupture and contralateral ACL injuries. However, it is important to note that this review had inherent limitations, such as a shorter follow-up duration that might not have been sufficient to reveal clinical benefits, particularly if the event rates were lower than expected. Consequently, the review yielded broad 95% confidence intervals, resulting in a lack of statistically significant differences. Moreover, additional studies have since been published, which should be included in subsequent meta-analyses to provide updated results. Considering the musculoskeletal disease remains the disturbing issues for people worldwide (*Li et al., 2019*; *Wang et al., 2021*; *Wei et al., 2022a*; *Wei et al., 2022b*), we therefore conducted a systematic review and meta-analysis to evaluate the association between passing an RTS test and the risk of knee injury, secondary ACL injury, contralateral ACL injury, or graft rupture.

The summary results indicated that passing an RTS test by athletes was not associated with an increased risk of knee injury, secondary ACL injury, and contralateral ACL injury. These findings were consistent with the ones of a previous meta-analyses conducted by *Losciale et al. (2019)* and *Webster & Hewett (2019)*. However, it is important to note that while most of the included studies had similar conclusions, several other studies reported inconsistent results (*Losciale et al., 2019*; *Webster & Hewett, 2019*). For example, investigations performed by *Grindem et al. (2016)* and *Kyritsis et al. (2016)* found that passing an RTS test was associated with a lower risk of knee injury. This could be explained by the fact that individuals who did not pass the RTS test had larger kinematic and kinetic asymmetries between limbs, and a gait strategy similar to early-stage athletes was used in these earlier studies. Therefore, it is important to rigorously apply the RTS test in relation to known measures of biomechanical impairments (*Di Stasi & Snyder-Mackler, 2012*). On the other hand, a study conducted by *Sousa et al. (2017)* found that athletes who passed an RTS test had an increased risk of knee injury and secondary ACL injury. This result could be explained by the excessive risk of contralateral ACL injury in individuals who passed the RTS test. Sensitivity analysis also suggested that the risk of contralateral ACL injury might have been increased in patients who passed the RTS test. However, passing an RTS test was associated with a reduced risk of graft rupture. This could be attributed to the increased loading of the contralateral limb at the time of return to sport and beyond. It is worth noting that this increased loading of the contralateral limb may also contribute to the increased risk of contralateral injury post-release to return to play (*Paterno et al., 2007*; *Paterno et al., 2012*). Notably, the risk of graft rupture differed between athletes who passed the RTS test and those who did not. This suggests that the risk of graft rupture is lower in individuals who passed an RTS test (*Kyritsis et al., 2016*).

Interestingly, we noted a protective role of passing an RTS test in Europe, whereas there was a harmful effect of passing the RTS test in the USA. One potential reason for this discrepancy could be that the criteria for passing the RTS test in Europe were stricter than those in the USA. Additionally, we observed that passing an RTS test was associated with an increased risk of contralateral ACL injury when the pooled studies had a male proportion <60.0%. This finding may be related to the vulnerability of female athletes (*Prodromos et al., 2007*). Furthermore, passing the RTS test was associated with a reduced risk of graft

rupture when the mean age of patients <25.0 years. This result suggests that the RTS test could be used to identify a specific population at high risk for graft rupture. Finally, the risk of knee injury and contralateral ACL injury in athletes passing the RTS test was observed in pooled studies with a low quality. Therefore, these conclusions need to be further verified through prospective studies to account for uncontrolled biases.

Several limitations of this study should be acknowledged: (1) both prospective and retrospective studies were included, which introduced the possibility of inevitable selection and recall biases; (2) the analysis of this study was based on crude data, and potential confounders were not adjusted for; (3) there was variation in the RTS test battery, RTS test time, and the pass rate across the included studies, which could impact the prognosis of athletes after ACL injury; (4) publication bias was inevitable due to the analysis of published articles; and (5) the analysis in this study utilized pooled data, limiting its potential for comprehensiveness.

## CONCLUSIONS

The findings of this analysis suggest that athletes who pass the RTS test do not have an increased risk of knee injury, secondary ACL injury, and contralateral ACL injury. Additionally, passing an RTS test is associated with a reduced risk of graft rupture. Furthermore, the protective role of passing the RTS test is more evident in Europe, whereas it may be associated with a poor prognosis in the USA. These findings should be verified in further large-scale prospective studies.

### Funding
The study was supported by the Sichuan Provincial Science and Technology Plan Project of Traditional Chinese Medicine Orthopedics and Sports Rehabilitation Clinical Medical Research Center of Sichuan Province (Grant No. 2019YFS0541) and Sichuan Provincial Science and Technology Plan Project of Research on Key Techniques for Prevention and Rehabilitation of Sports Injuries in high performance athletes (Grant No. 2020ZHYZ0005). The funders had no role in study design, data collection and analysis, decision to publish, or preparation of the manuscript.

### Grant Disclosures
The following grant information was disclosed by the authors:
Sichuan Provincial Science and Technology Plan Project of Traditional Chinese Medicine Orthopedics and Sports Rehabilitation Clinical Medical Research Center of Sichuan Province: 2019YFS0541.
Sichuan Provincial Science and Technology Plan Project of Research on Key Techniques for Prevention and Rehabilitation of Sports Injuries in high performance athletes: 2020ZHYZ0005.

## Competing Interests

The authors declare there are no competing interests.

## Author Contributions

- Wenqi Zhou conceived and designed the experiments, performed the experiments, analyzed the data, prepared figures and/or tables, authored or reviewed drafts of the article, and approved the final draft.
- Xihui Liu performed the experiments, analyzed the data, authored or reviewed drafts of the article, and approved the final draft.
- Qiaomei Hong performed the experiments, authored or reviewed drafts of the article, and approved the final draft.
- Jingping Wang performed the experiments, authored or reviewed drafts of the article, and approved the final draft.
- Xiaobing Luo conceived and designed the experiments, performed the experiments, prepared figures and/or tables, authored or reviewed drafts of the article, and approved the final draft.

## Data Availability

This is a meta-analysis.

## Supplemental Information

Supplemental information for this article can be found online at http://dx.doi.org/10.7717/peerj.17279#supplemental-information.

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
