# Peer review of "Association between passing return-to-sport testing and re-injury risk in patients after anterior cruciate ligament reconstruction surgery: a systematic review and meta-analysis"

_PeerJ, doi:10.7717/peerj.17279_

## Round 0.1 · original submission · Major Revisions

Please carefully read the comments and suggestions from the reviewers and provide your point-by-point responses.

**Language Note:** The review process has identified that the English language must be improved. PeerJ can provide language editing services - please contact us at copyediting@peerj.com for pricing (be sure to provide your manuscript number and title). Alternatively, you should make your own arrangements to improve the language quality and provide details in your response letter. – PeerJ Staff

·

Basic reporting

.

Experimental design

.

Validity of the findings

.

Additional comments

1. Please upload a retraction table of all raw data, though all data were extracted from public database
2. How the authors explain the clinical meaning of your finding?
3. Please provide a detailed search strategy for your pubmed and embase database.
4. what effect model you use? fixed effect or random effect model for forest map?
5. The references shall be updated with more recent literature.

·

Basic reporting

In this manuscript, Zhou et al. The association between passing return1to1sport testing and
prognosis for patients after anterior cruciate ligament reconstruction surgery in a meta-analysis way. Overall, this manuscript is well-done and have raised much strengths whereas some issues must be addressed before publishing.

Experimental design

no comment

Validity of the findings

no comment

Additional comments

1. The language is chaotic and a native speaker is needed before publish.
2. Similar articles are published and can you clarify the innovations?
3. The description of methods and results in P4 abstract is inconsistent with that in P6 abstract.
4. It is mentioned in Lines 66-67 that there is no restriction on language when searching literature, but this paper didn’t search Chinese literature databases such as CNKI.
5. Please cite more appropriate recent literature for Section Introduction.
6. For Section Methods, please refer to the methodology of the following articles and cite them: PMID: 35032305, 37703284, 34321360
7. The contents of Lines 230-234 should belong to the conclusion,not discussion.
8. Most references are old.
9. Can authors beautify the figure 1-5?
10. Please revise the fig 1 according to the PRISMA guideline.

·

Basic reporting

Please take a closer look at the professional English being used:
General comments:

English phrasing - would recommend reading by a fluent English-speaking reviewer.
In general, I found that your introduction could use more recent insights on the problem for RTS and reinjuries. Please take my following suggestions under consideration.

Within your discussion there could be an opening for discussion concluding the difference in RTS testing batteries and the more sensitive tests are found within biomechanical analysis?

Comments within the article itself:

Line 41-44
I think you can find more sources to make your point. My question would be why you only mention female athletes.

Line 46-47
Could you build this sentence with recent literature that ACL reconstruction results in not performing on their former level of participation?

Line 55
My suggestion would be also to use studies from M Buckthorpe on ACLR rehabilitation.

Line 57
Ardern (2016) and Buckthorpe describe the used RTS criteria for decision making for RTP

Line 59
What inconsistent roles do you mean/ did you find?

Line 62
Don’t you mean that the citeria are not completely clarified? And not just the role of testing?

Line 84
Again: do you examine the role of testing or the resulting criteria mentioned?

Line 94-96 conflicts with line 118-119

Line 140
You mention ‘irrelevant studies’: could you be more specific?

Line 217
Only graft rupture? Or also other injuries you mentioned before?

Line 237
The findings are also in line with research was conducted in our facility by Wouter Welling ea ‘ passing return to sports tests after ACL reconstruction is associated with greater likelihood for return to sport but fail to identify second injury risk’ 2020

Line 238-245
I do not completely understand what you are trying to say here referring to the study of Kyritsis et al. While their conclusion was: “ Athletes who did not meet the discharge criteria before returning to professional sport had a four times greater risk of sustaining an ACL graft rupture compared with those who met all six RTS criteria. In addition, hamstring to quadriceps strength ratio deficits were associated with an increased risk of an ACL graft rupture.” How is this inconsistent to your findings?

Experimental design

no comments on the experimental design

Validity of the findings

no comments

Additional comments

no further comments

Reviewer 4 ·

Basic reporting

he English of the article is good and the grammar is appropriate and sufficient. Literature reference could be improved in introduction and discussion. Results and explanation must be more consistent and relevant

Experimental design

It is an article written in accordance with the scope of the journal. Basic working knowledge of the subject being reviewed is sufficient T
introduction: the authors should bring more detail ad explain why RTS is controversy as the authors wrote in abstract.Research question is clearly stated. The methodology is aptly described but the authors should bring more details about your selection criteria. The results of the article are explained in an appropriate way with text and tables. Discussion and conclusion:It is clearly and appropriately stated by the authors but discussion must be more accuracy and relevant

Validity of the findings

findings must be more accuracy, consistent and relevant. The discussion lack of accuracy about RTS testing and the authors didn't bring the results form the others study. Unfortunately, the reader couldn't compare the results of this study and the others. It's very important that the reader can make his own idea. I think the studied population is too heterogenous and could influence the results. The authors must do any stratifications to be more clearly.

Annotated reviews are not available for download in order to protect the identity of reviewers who chose to remain anonymous.

---

## Round 0.2 · Minor Revisions

Please address the comments from reviewers.

·

Basic reporting

There are sufficient literature references to prove that this is a meaningful study. Whether patients can safely return to exercise after ACL reconstruction has been a hot research topic in recent years. And The structure of this paper conforms to the normal structure of meta-analysis, there are sufficient figures and tables to prove the conclusion.

Experimental design

The method of this article is reasonable, and the methods are described in sufficient detail but there is a problem that the database searched by the author does not contain PubMed, PubMed may contain literature that has not been included.

Validity of the findings

All characteristics of all included studies have been provided, There are appropriate forest figures, subgroup analyses, and publication bias analyses. This study found that passing the RTS test was not associated with the risk of subsequent knee injury, secondary ACL injury, and contralateral ACL injury, while it was associated with a lower risk of graft rupture, this conclusion can guide athletes to return to sport after ACLR.

Additional comments

no comment

·

Basic reporting

no comments

Experimental design

no comment

Validity of the findings

minor comments, see attachment

Additional comments

Congratulations on your work so far. Please take my last considerations into account. In my opinion this will improve the readability of your work.
Yours sincerely

---

## Round 0.3 · accepted · Accept

The authors have addressed the concerns.